# Data from the PASSI d’Argento Surveillance System on Difficulties Met by Older Adults in Accessing Health Services in Italy as Major Risk Factor to Health Outcomes

**DOI:** 10.3390/ijerph191610340

**Published:** 2022-08-19

**Authors:** Benedetta Contoli, Valentina Possenti, Rosaria Gallo, Valentina Minardi, Maria Masocco

**Affiliations:** 1National Centre for Disease Prevention and Health Promotion, Istituto Superiore di Sanità, 00161 Rome, Italy; 2Primary Healthcare Unit, Health District 9, Local Health Unit Roma 2, 00159 Rome, Italy; 3PhD Course Advances in Infectious Diseases, Microbiology, Legal Medicine and Public Health Sciences, Sapienza University of Rome, 00185 Rome, Italy

**Keywords:** age-friendly health systems, older adults, surveillance system, health service accessibility, prevention, Italy

## Abstract

(1) Age-friendly health systems ensure access to quality healthcare services to all people, especially older adults. (2) We used data on elderly population collected from 2016 to 2019 by the Italian ongoing surveillance system PASSI d’Argento to analyze the prevalence and associations between accessing health services and modifiable risk factors included in the 25 × 25 strategy for the burden of noncommunicable diseases with health outcomes. (3) Chronic diseases and hospitalization as descriptors of health status showed that the elderly perceived as having poor access to care and prevention incurred a higher risk of hospitalization. The association between difficulties in accessing health services and hospitalization was always the highest in terms of the adjusted prevalence ratio (aPR), regardless of the other behavioral risk factors considered, controlling each model with sociodemographic conditions. Elderly hospitalized at least once for two days or more in the last 12 months had greater risk to have problems in accessing health services, whereas the model included health conditions such as obesity (aPR = 1.95 95% CI 1.75–2.17), smoking (aPR = 1.95 95% CI 1.76–2.16), alcohol use (aPR = 1.93 95% CI 1.73–2.14), hypertension (aPR = 1.92 95% CI 1.73–2.13) and diabetes (aPR = 1.91 95% CI 1.73–2.12). (4) Health policies should encompass socio-economic and living environment barriers which prevent access to care among older adults.

## 1. Introduction

Over the last two decades, although the lengthening of the average lifespan and the constant increase in population aging represent a great success achieved by medicine and science overall, they have become top priorities in global health policies and strategic plans, which aim to identify appropriate actions and interventions both to tackle problems related to advancing health, and concerning the development of well-being and supportive environments for elderly people. As early as when the Political Declaration and Madrid International Plan of Action on Aging were approved in 2002, formal explication has been retrievable about either ensuring equal access to basic and appropriate social services for older persons, especially in rural and remote areas, or the importance of older persons having physical access to services, and also being involved in planning and delivering. Indeed, as people age, their access to public transport, entertainment facilities, day centers, doctors, hospitals, care homes and support services becomes increasingly important. The necessity to provide older persons with the same access to preventive and curative care and rehabilitation as other groups has been recognized, and, at the same time, the need for health services to be designed to meet the special needs of the older population [1]. The aging phenomenon does indeed raise new challenges, in particular to socio-health systems, as well as to the societal arrangement overall. Effective strategies to win against these grand challenges are in fact well known, so much so that several countries have put them into practice through joint efforts provided at different levels, supplying integrated and equitable forms of care, the latter even including service accessibility and support to caregivers on the one hand, but also fighting against poverty and the promotion of the competent roles that the elderly may have in society on the other hand [2,3,4,5,6].

Additionally, the process of progressive population aging is strongly associated with the constant growth of noncommunicable diseases (NCDs) [7], and is linked to other health-related issues: the increasing frequency of non-self-sufficient elderly, the growing prevalence of comorbidities, worsening of frailty and disability conditions [8]. This is why actions and interventions aimed at ensuring a good quality of life and decreasing the risk for either frailty or disability in the elderly are crucial within strategies and policies for NCD prevention. The elaboration of the framework for “*Active and Healthy Aging*” (AHA) then occurred as a further contribution by the World Health Organization (WHO) to the Second United Nations World Assembly on Aging in Madrid in 2002 [9]. Sustained on three pillars (participation, health and security), this paradigm is the outcome resulting from different determinants that allow for the identification of particular profiles, which might be more at risk or, conversely, more favorable to age actively [10]. Furthermore, in 2011, the WHO 25 × 25 initiative followed, which was a strategy to reach a global target of a 25% relative reduction in mortality from NCDs by the year 2025, through lowering interventions of six main modifiable risk factors: −10% in harmful alcohol consumption, −30% in tobacco smoking, −30% in salt intake, −25% in hypertension and no increase in the prevalence of obesity and diabetes and −10% in sedentary behaviors [11]. As already highlighted by the WHO in its Global Action Plan for Healthy Lives and Well-being for All [12], the Lifepath study showed that socioeconomic determinants should be targeted using local and global health strategies in addition to the 25 × 25 factors, because they serve as the “cornerstones” for the achievement of the Sustainable Development Goals (SDGs) related to health [13].

Aging does actually represent a relevant issue in Italy, where one out of four individuals is aged 65 years old and over. By 2050, they are projected to increase by nearly five million people and, given declining fertility rates and increased life expectancy, thus, would become approximately 35% of the general population [14]. Among the several health-related behaviors under study, access to basic healthcare and social services is one of the aspects investigated by PASSI d’Argento (PdA) [15], an ongoing national behavioral risk factor surveillance system of Italian residents aged 65 years and over. PdA, which began as a series of periodic cross-sectional surveys conducted between 2009 and 2015, became a surveillance system in 2016. Since then, it has been continuously collecting information about quality of life, modifiable risk factors, participation and engagement in society, independent living, health conditions, safety and living environments [16]. Other PdA topics under investigation are perceived health and service coverage, life satisfaction and, in accordance with the AHA concept, the “measurement” of the contribution that elderly people offer to society, supporting their family and community by maintaining good health and high-quality relations [17].

We used data from PdA collected from 2016 to 2019 to measure the prevalence and detect the association between accessing health services and the 25 × 25 strategies’ risk factors (at-risk alcohol consumption, sedentary behavior, current tobacco use, raised blood pressure, diabetes and obesity) with three health outcomes, as having been hospitalized once or more in the last 12 months and suffering from at least one chronic disease. Indicators of salt or sodium intake were not analyzed, because they are not topics under investigation in PdA.

## 2. Materials and Methods

### 2.1. Sampling and Data Collection Methods

The PdA surveillance system, dedicated to people over 65 years of age residing in Italy, is centrally coordinated by the Italian National Institute of Health (Istituto Superiore di Sanità, Rome, Italy) in collaboration with the regional health authorities.

Since 2016, PdA has been continuously gathering information on the elderly population living in Italy, including their quality of life and health conditions, behavioral risk factors, participation in society, perceived health and service coverage, independent living, safety and living environments and others.

Specifically trained personnel from social and health services collected such data by telephone or face to face interview, using a standardized questionnaire that included more than 80 closed-ended questions. Study variables included sociodemographic and socioeconomic conditions (gender, age, educational qualifications, economic conditions, marital status, family composition, nationality) and difficulty in accessing health services, that is, accessing information for tackling health inequalities.

The sample was randomly selected from the local health unit (LHU) list of beneficiaries of health services in the participating regions, stratified by sex and age (65–74, 75–84 and ≥85). Exclusion criteria were as follows: being a resident of or having a permanent address in another region, not having a contact telephone number, being currently hospitalized or in long-term care, currently living in a nursing home or prison, being deceased and, finally, not being a speaker of Italian. Approximately 13,000 older people were interviewed every year. However, further methodological descriptions are available elsewhere [18,19,20].

In the time period 2016–2019, all the Italian regions, with the exception of Lombardy and Valle d’Aosta, developed the PdA surveillance system, achieving an amount of 56,352 records collected, with a response rate of over 86% and a refusal rate equal to 10% [21]. The interviewees were assigned a proxy (a relative or caretaker) if, at the beginning of the interview, they were not able to answer at least three out of five questions of the short memory test, derived from the Mini-Mental State Examination [22]. Indeed, from the overall sample, all the respondents who needed a proxy to be interviewed were excluded from the analysis. The final sample under analysis included 45,514 older individuals.

### 2.2. Study Variables and Definitions

The independent variables considered in our analyses included the following:Age: 65–74, 75–84, 85+ years;Gender: male, female;Education: low (none or elementary school), high (middle school or higher);Economic difficulties (in making ends meet with the available household financial resources): none, some, many;Geographic area: north, center, south and major islands;Living alone: yes, no.

Difficulties in accessing health services, LHU and general practitioners (GPs):Respondents reporting difficulties to reach the LHU and GPs: some, many;Behavioral risk factors (one for each model): tobacco smoking, higher-risk alcohol consumption, physical inactivity according to the WHO recommendations (using the physical activity scale for Elderly, PASE [23,24]), hypertension, diabetes, obesity (body mass index, BMI > 30).

The health outcomes:One or more hospitalizations (lasting two days or more) in the last 12 months before the interview: yes, no;Chronic diseases (self-reported presence of diabetes, kidney failure, chronic bronchitis, emphysema, asthma, respiratory insufficiency, stroke, myocardial infarction, coronary and other heart diseases, tumors, chronic liver disease or cirrhosis): none, at least one.

### 2.3. Statistical Analysis

Firstly, we estimated the prevalence of elderly who received a diagnosis of at least one or two chronic diseases in their lifetime and had at least one hospitalization within the 12 months before the interview. Data were also analyzed by subgroups, considering sociodemographic characteristics and health risk factors. Prevalence data were computed with confidence intervals at 95% (95% CI).

We performed Poisson regression analyses [25] for each specific health outcome or behavioral risk factor (sedentariness, smoking, at-risk alcohol consumption, diabetes, hypertension, obesity and difficulties in accessing health services) controlling each model using sociodemographic variables such as age, gender, education level, economic difficulties and geographic area of residence. Adjusted prevalence ratios (aPR) were presented with corresponding 95% CI and *p*-value.

In the second phase, we analyzed the characteristics of the elderly having problems in accessing health services, where the aPRs were shown to be associated with the sociodemographic variables.

The software STATA, version 16.0 (Stata Corp, College Station, TX, USA) supported all data analyses [26].

## 3. Results

### 3.1. Description of PdA 2016–2019 Respondents by Sociodemographic and Health Conditions in Association with the Presence of Chronic Diseases and Hospitalization

The analyses considered 45,514 out of the 56,352 records collected on individuals aged 65 and older by the PDA from 2016 to 2019. Table 1 reports the sociodemographic and health characteristics of these interviewees.

Over half (57%) of the respondents belonged to the 65–74 years age group, and there were more females than males (56% versus 44%).

Almost half of the respondents reported financial difficulties, 43% had a low level of education and 23% referred to problems in accessing health services.

Although 18% had received a diagnosis of diabetes, 14% was obese, 59% had hypertension, 40% had a sedentary lifestyle, 21% reached an excessive alcohol intake, 11% smoked and 28% was represented by former smokers.

The other results shown in Table 1 were based on three groups: elderly with one or more chronic diseases (56%), with two or more chronic diseases (21%) and those who reported one or more hospitalizations of two days or longer in the 12 months before the interview (13%).

The three groups presented similar demographic characteristics, such as age, gender and living status. Instead, the prevalence of respondents with economic difficulties, a low educational level and residing in the south of Italy reached a greater percentage in the group of elderly with comorbidities (respectively, 62%, 51% and 45%).

The prevalence of people with problems in accessing health services was higher in the three groups than the average of the sample (23%), with the highest value in the subgroup of ones with several comorbidities (39%).

We observed similar unhealthy behaviors among the three groups with the prevalence of a sedentary lifestyle and obese people being significantly higher than that found in the sample.

A diagnosis of diabetes was reported more frequently by elderly with two or more chronic diseases (27%).

### 3.2. Association between Health Conditions and Access to Health Services with Presence of Chronic Diseases and Hospitalization

A total of 18 models, 6 for each of the three outcomes, was elaborated. Results of each multivariate regression model are shown in Table 2.

Each model was adjusted for sociodemographic variables (gender, age, economic difficulties, level of education, geographic area of residence and current living status). A multivariate regression analysis confirmed that by controlling for gender, age, geographic area, current living status, educational level and economic conditions in each model, the health profile and behavioral risk factors, except at-risk alcohol drinking, were significantly associated with the presence of chronic diseases or hospitalization.

The most important association was having difficulties in accessing health services such as the LHU or GPs. At-risk alcohol use (such as two or more units per day) was never statistically significant.

By comparing the models, elderly people with two or more chronic diseases incurred a higher risk of developing diabetes (aPR = 1.45% 95% CI 1.31–1.55) or hypertension (aPR = 1.50% 95% CI 1.39–1.63), to be obese (aPR = 1.38 95% CI 1.25–1.52), to smoke (aPR = 1.43 95% CI 1.26–1.63) or to have smoked formerly (aPR = 1.37 95% CI 1.26–1.49). Additionally, in each model, older people with comorbidities were always at risk of having problems in accessing health services. Among the elderly who had at least one hospitalization in the 12 months prior to the interview, a greater risk of experiencing difficulties in accessing health services occurred when the model included health conditions such as obesity (aPR = 1.95 95% CI 1.75–2.17) or tobacco smoking (aPR = 1.95 95% CI 1.76–2.16), alcohol use (aPR = 1.93 95% CI 1.73–2.14), hypertension (aPR = 1.92 95% CI 1.73–2.13) or diabetes (aPR = 1.91 95% CI 1.73–2.12). However, the association between difficulties in accessing health services and hospitalization resulted in always being highest in terms of aPR, regardless of the other risk factors considered.

### 3.3. Which Is the Profile of Elderly Reporting Difficulties in Accessing Health Services?

Among the 45,514 older adults interviewed between 2016 and 2019 included in this study, 57% reported difficulties in accessing health services. The characteristics and the results of the multivariate analyses of risk factors for this sample are shown in Table 3.

Elderly people having difficulties in accessing health services were more likely to be female (aPR = 1.69 95% CI 1.58–1.82) and aged 75 years or older (75–84: aPR 1.78 95% CI 1.67–1.91; 85+: aPR = 2.92 95% CI 2.70–3.17), have some (aPR = 1.73 95% CI 1.61–1.87) or many (aPR = 2.15 95% CI 1.98–2.35) economic difficulties, a low level of education (aPR = 1.39 95% CI (1.30–1.48) and residing in the south (aPR = 1.86 95% CI 1.71–2.01) or center of Italy (aPR = 1.37 95% CI 1.25–1.51).

## 4. Discussion

We used a large population sample, representative of age and gender, of elderly residents in Italy to compare the association of having difficulties accessing health services with three selected health outcomes, and the association with the 25 × 25 risk factors, as posed by the WHO strategy to reach a 25% relative reduction in premature mortality from NCDs by the year 2025. In our study, health status was expressed in terms of suffering from chronic diseases and having been hospitalized at least once in the last 12 months. As others showed [27,28], we also confirmed that, indeed, elderly people who were perceived as having poor access to care and prevention were at greater risk of hospitalization. Such findings from the PdA behavioral surveillance addressing the health service accessibility as being a transversal issue fell within the principles of the age-friendly health system (AFHS) [29] framework. The AFHS in fact aims to improve the experience of care for the elderly through the 4Ms paradigm, an evidence-based approach to care planning that emphasizes *what matters* most to the older people, *mentation, mobility* and *medication*, having a direct bearing on the other three elements [30,31]. From this perspective, it is relevant to acquire the understanding that problems and difficulties in accessing health services have an important influence on and strong association with health outcomes, no less relevant than health risk behaviors and social or financial conditions. The base concept of the 4Ms approach for the AFHS framework is one of the intended global goals established by the WHO to align health services to the needs of the elderly, fully corresponding to the idea of expertly designed health systems to coordinate all care as we age and being able to equitably address the environments where especially older people live, learn, work and play [29]. The association between having difficulties in accessing health services and social determinants of health is well known [32,33]. Among its potential effects on health, in fact, a low care access even includes the poor management of chronic disease, an increased burden due to preventable diseases and disability and premature death. It is why access to comprehensive and quality healthcare services is important for promoting and maintaining health, preventing and managing diseases, reducing unnecessary disabilities and premature death and achieving health equity for all, especially for vulnerable people such as the elderly. Facilitating citizen access to services would then allow for the implementation of specific actions on the main risk factors of chronic–degenerative diseases, which are of great epidemiological importance.

The PdA investigation on how much older adults living in Italy think that health services are accessible represents a synthetic measure of the five characteristics for healthcare access (approachability, acceptability, availability and accommodation, affordability and appropriateness) as described in the Levesque conceptual framework [34]. These results provide information of paramount importance to understanding how access to care and the prevention of chronic diseases, frailty and disability for people advancing with age can be improved. Given that healthcare should be provided when people need it, through a balanced geographical distribution of healthcare facilities, a blended critical social perspective already revealed the overlapping and complex determinants that affect primary healthcare utilization, concluding with the importance of situating healthcare access in sociocultural structures. As we also found, access to healthcare affects the self-assessed health and quality of life of the elderly with chronic diseases, especially in patients with poor health, though differently for urban and rural patients. Indeed, PdA indicators resulted in being flexible tools for providing accurate data on this extremely complex matter, as also defined by health-related SDGs [18]. Policy actions targeted at especially underserved subgroups of older people, such as vulnerable and rural populations, should give priority to reducing barriers in seeking health services, and to contrast the benefits of rural and urban life, as the former is often perceived as a cheaper and safer environment and the latter tends to provide a wider range of services and amenities at far greater proximity [35,36]. Furthermore, those living alone might represent even more particularly vulnerable elderly and, to them, an important indicator for health services accessibility was the availability of information regarding the services, that is, depending on both the professional’s good will and the older person’s own financial resources [37,38]. In Italy, each person is assigned a GP, and many of the elderly respondents had longstanding relationships with these physicians and trust in their advice. Moreover, centralized records are kept in each LHU of the physician to whom each person is assigned; thus, this could be used as a way of identifying those who might benefit from forms of tailored care such as individual letters or calls from their GPs. Although the Italian health system is mostly grounded on a public basis, direct and indirect cost-related issues may represent social factors hindering access to care, increasing the seniors’ precariousness. Indeed, if older persons are likely to experience even financial and legal barriers to healthcare services, costs should never prevent people from receiving the healthcare they need [39].

Our study had some limitations, mainly due to the fact that the PdA data were self-reported and collected by a telephone interview; they may, therefore, be biased somehow. In the end, this research had major strengths, because the PdA is an ongoing surveillance system and can monitor how the distribution of elderly reporting difficulties in accessing health services changes over time, even under emergency scenarios such as the COVID-19 pandemic [40].

## 5. Conclusions

The present research provided further updated evidence about what the consistent burden equity in accessing health services means in terms of health outcomes, at least as much as behavioral risk factors and socioeconomic conditions are concerned. Understanding the extent to which older people may have difficulties in accessing health services and how they are associated with health outcomes is important for the development of elderly tailored interventions at individual and population levels. The perspective for better aging was intended in a community scenario: on the one hand, staying socially active and creative, providing access to public and commercial services, thus, improving quality of life and reducing social isolation; on the other hand, providing accessible and effective healthcare services that promote the early detection of diseases and helping older people to maintain their health and capacity to live independently, while ensuring that healthcare spending remains under control. To implement such complex interventions for healthier, stronger and more resilient older individuals, monitoring the aging population in how their health status and modifiable factors change over time does actually represent a key step.

## Figures and Tables

**Table 1 ijerph-19-10340-t001:** Sociodemographic and health characteristics of elderly: overall, by presence of chronic diseases and hospitalization and relative confidence intervals at 95% (95% CI). Passi d’Argento, 2016–2019 (*n* = 45,514 out of 56,352).

		Distribution of Sample		At Least One Chronic Disease		At Least Two Chronic Diseases		At Least One Hospitalization ^1^	
		N = 45,514		N = 25,883		N = 9765		N = 6455	
		%	CI 95%	%	CI 95%	%	CI 95%	%	CI 95%
Total				57	(55.9–57.5)	21	(20.7–22.0)	13	(12.7–13.7)
Gender	Male	44	(43.1–44.4)	46	(44.8–46.9)	46	(44.3–48.3)	45	(43.1–47.3)
	Female	56	(55.6–56.9)	54	(53.1–55.2)	54	(51.7–55.7)	55	(52.7–56.9)
Age group	65–74	57	(56.5–57.8)	51	(49.8–51.8)	45	(42.7–46.7)	50	(47.8–51.9)
	75–84	35	(34.2–35.4)	39	(38.2–40.2)	43	(41.3–45.2)	39	(37.4–41.4)
	85+	8	(7–8–8.4)	10	(9.4–10.6)	12	(11.0–13.2)	11	(9.6–12-1)
Educational level	Low	43	(42.4–43.9)	47	(45.5–47.7)	51	(49.0–53.1)	46	(43.7–47.9)
	High	57	(56.1–57.6)	53	(52.3–54.5)	49	(46.9–51.0)	54	(52.1–56.3)
Economic difficulties	Yes	48	(46.8–48.4)	54	(53.2–55.4)	62	(60.0–63.8)	46	(43.8–48.0)
	No	52	(51.7–53.3)	46	(44.62–46.8)	38	36.2–40.1	54	(52.0–56.2)
Difficulties in accessing 1+ health services	Yes	23	(22.3–23.6)	30	(29.1–31.2)	39	(36.8–40.8)	36	(34.2–38.5)
	No	77	(76.4–77.7)	70	(68.8–70.9)	61	(59.2–63.2)	64	(61.5–65.8)
Living alone	Yes	22	(20.9–22.2)	22	(20.7–22.5)	22	(20.4–23.7)	23	(20.9–24.3)
	No	79	(77.9–79.1)	78	(77.5–79.1)	78	(76.3–79.6)	77	(75.7–79.1)
Geographic area of residence	North	39	(38.2–39.7)	37	(36.4–38.6)	35	(32.8–36.8)	40	(37.8–41.9)
	Center	20	(20.1–20.8)	20	(19.1–20.6)	20	(18.4–21.5)	21	(19.0–22.2)
	South	41	(39.9–41.2)	43	(41.7–43.7)	45	(43.4–47.3)	40	(37.7–41.7)
Sedentary behavior	Yes	40	(39.3–40.8)	44	(42.8–45.2)	47	(44.8–49.4)	54	(54.3–60.3)
	No	60	(59.2–60.8)	56	(54.8–57.2)	53	(50.6–55.2)	46	(38.7–45.7)
Tobacco smoking	Yes	11	(10.5–11.4)	12	(10.9–12.4)	12	(10.9–14.3)	11	(9.7–13.3)
	Former smoker	28	(27.4–28.7)	31	(29.8–31.7)	32	(29.8–33.4)	31	(29.6–33.3)
	No	61	(60.3–61.8)	58	(56.6–58.7)	56	(53.9–57.9)	57	(55.2–59.4)
At-risk alcohol consumption	Yes	21	(20.0–21.3)	21	(19.7–21.5)	19	(17.0–20.5)	19	(17.5–21.4)
	No	79	(78.8–80.0)	80	(78.6–80.4)	81	(79.5–83.0)	81	(78.6–82.5)
Diabetes	Yes	18	(17.9–19.1)	21	(20.6–22.4)	27	(25.1–29.0)	23	(20.6–24.6)
	No	82	(80.9–82.2)	79	(77.6–79.4)	73	(71.0–74.9)	77	(75.4–79.4)
Hypertension	Yes	59	(58.1–59.6)	65	(64.2–66.3)	65	(64.2–66.3)	63	(61.0–65.0)
	No	41	(40.4–41.9)	35	(33.7–35.8)	35	(33.7–35.8)	37	(35.0–39.0)
Obesity	Yes	14	(13.4–14.5)	16	(15.4–17.0)	19	(17.6–20.7)	17	(15.1–18.2)
	No	86	(85.5–86.6)	84	(83.0–84.6)	81	(79.3–82.4)	83	(81.8–84.9)

^1^ Duration of hospitalization: two or more days.

**Table 2 ijerph-19-10340-t002:** Adjusted prevalence ratios (aPR by Poisson regression model) and relative confidence intervals at 95% (95% CI), for health risk factors and access to health services associated with health outcomes (presence of chronic diseases, hospitalization). Passi d’Argento, 2016–2019 (*n* = 45,514 out of 56,352).

		Health Outcomes					
		At Least One Chronic Disease		At Least Two Chronic Diseases		At Least One Hospitalization ^1^	
		aPR	CI 95%	aPR	CI 95%	aPR	CI 95%
	Sedentary behavior	1.08	(1.04–1.12)	1.10	(1.01–1.21)	1.20	(1.09–1.31)
Models	Difficulties in accessing 1+ health services	1.26	(1.21–1.32)	1.60	(1.43–1.79)	1.62	(1.43–1.83)
	Tobacco smoking						
	Former smoker (vs. never smoker)	1.20	(1.14–1.26)	1.37	(1.26–1.49)	1.24	(1.13–1.36)
	Current smoker (vs. never smoker)	1.20	(1.16–1.25)	1.43	(1.26–1.63)	1.19	(1.02–1.38)
	Difficulties in accessing 1+ health services	1.33	(1.28–1.38)	1.78	(1.61–1.95)	1.95	(1.76–2.16)
	At-risk alcohol consumption (vs. no drinking)	1.01	(0.97–1.05)	0.93	(0.83–1.03)	0.94	(0.84–1.06)
	Difficulties in accessing 1+ health services	1.32	(1.27–1.37)	1.75	(1.59–1.93)	1.93	(1.73–2.14)
	Diabetes	1.13	(1.08–1.17)	1.43	(1.31–1.55)	1.19	(1.08–1.31)
	Difficulties in accessing 1+ health services	1.31	(1.26–1.36)	1.70	(1.55–1.87)	1.91	(1.73–2.12)
	Hypertension	1.25	(1.21–1.29)	1.50	(1.39–1.63)	1.12	(1.03–1.21)
	Difficulties in accessing 1+ health services	1.30	(1.25–1.35)	1.70	(1.55–1.87)	1.92	(1.73–2.13)
	Obesity	1.16	(1.11–1.21)	1.38	(1.25–1.52)	1.19	(1.07–1.32)
	Difficulties in accessing 1+ health services	1.31	(1.26–1.37)	1.74	(1.57–1.93)	1.95	(1.75–2.17)

^1^ Duration of hospitalization: two or more days.

**Table 3 ijerph-19-10340-t003:** Characteristics of elderly reporting difficulties in accessing health services and relative confidence intervals at 95% (95% CI). Passi d’Argento, 2016–2019 (*n* = 9383).

		Distribution of Sample		Elderly Reporting Difficulties in Accessing Health Services			
		N = 45,514					
		%	CI 95%	%	CI 95%	aPR	CI 95%
Total				57	(55.9–57.5)		
Gender	Male	44	(43.1–44.4)	29	(27.2–30.2)	1	-
	Female	56	(55.6–56.9)	71	(69.8–72.8)	1.692	(1.575–1.818)
Age group	65–74	57	(56.5–57.8)	38	(36.4–39.4)	1	-
	75–84	35	(34.2–35.4)	45	(43.7–46.6)	1.782	(1.665–1.906)
	85+	8	(7.8–8.4)	17	(16.1–18.0)	2.923	(2.699–3.165)
Educational level	Low	43	(42.4–43.9)	62	(60.0–63.4)	1.391	(1.303–1.484)
	High	57	(56.1–57.6)	38	(36.6–40.0)	1	-
Economic difficulties	Any	10	(9.7–10.6)	20	(18.6–21.29)	1	-
	Some	37	(36.6–38.2)	50	(48.3–51.8)	1.732	(1.605–1.869)
	Many	52	(51.7–53.3)	30	(28.5–31.6)	2.154	(1.975–2.350)
Living alone	Yes	22	(20.9–22.2)	25	(24.0–26.8)	0.966	(0.906–1.029)
	No	79	(77.9–79.1)	75	(73.2–76.0)	1	-
Geographic area of residence	North	39	(38.2–39.7)	25	(23.5–26.3)	1	-
	Center	20	(20.1–20.8)	19	(17.7–19.9)	1.374	(1.251–1.510)
	South	41	(39.9–41.2)	56	(54.8–57.8)	1.856	(1.711–2.013)
Safe neighborhood	Yes	86	(85.2–86.4)	80	(78.1–81.2)	1	-
	No	14	(13.6–14.8)	20	(18.8–21.9)	1.243	(1.146–1.347)
Structural problems in the house ^1^	Yes	62	(61.2–62.7)	72	(70.4–73.4)	1.127	(1.055–1.205)
	No	38	(37.3–38.9)	28	(26.6–29.6)	1	-

^1^ At least one housing problem.

## Data Availability

The data presented in this study are available on request from the corresponding author. The data are not publicly available due to restrictions, e.g., privacy or ethical.

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
