# Peer review of "Data from the PASSI d’Argento Surveillance System on Difficulties Met by Older Adults in Accessing Health Services in Italy as Major Risk Factor to Health Outcomes"

_ijerph, 2022, doi:10.3390/ijerph191610340_

Round 1

Reviewer 1 Report

This paper needs a thorough edit to fix English expression before it can be considered.  

For example:

the title doesn't make sense; 

there is confusion of causal and non causal relationships which appears to be a language issue;

on instead of one;

and other issues too numerous to list,

There are also structural errors, eg a random paragraph that does not belong in the Discussion and the abstract appears in the middle of the article. This may be an uploading problem, but suggests lack of care about quality of the paper.

Other matters

Attention is need to Table 3 heading and column headings.

Findings are lost in the tables, and thee repetition of details of results that are already reported in the tables does not help. Suggest use bold or italics to highlight the most significant relationships in the tables, especially Table 3.

Large chunks of the Discussion are about what the survey results can be used for, and not directly related to the results. They is insufficient consideration of the results in relationship to the research literature in the Discussion.

The Conclusion has little direct relevance to the results.

The paper can only be assessed when these matters are attended to.

Author Response

  • First, we thank the Reviewer 1 who suggested to improve the quality of the paper overall. Such those comments have been a spur to us to do better and - hopefully - to have increased the manuscript.
  • More in detail, in track-change modality, we provide a further option of title that we hope sounds more meaningful than the one we proposed with tentative abstract earlier and both were accepted by the Journal.
  • We hope that confusion between causal and non-causal relationship is clear because data presented are from the behavioural surveillance system, which is based on a cross-sectional study design. Thus, associations verified cannot be read in terms of causality.
  • We think that abstract appearing in the middle of the article was something due to fallacy, perhaps randomly occurring, of the platform. Anyway, we acted on several parts of the text, following the suggestion to improve the results’ presentation, organization of discussion and conclusions adjusting discourse linearity and consistency.
  • All three tables were elaborated further and data that are more significant highlighted as well.

Reviewer 2 Report

My congratulations to the authors for the well-written manuscript in which they investigated the association between difficulty in accessing health care services and three health outcomes as well as the association with the 25x25 risk factors. The manuscript is really interesting with a very topical subject, as WHO has recently proposed a strategy (the 25x25) to decrease the risk of mortality from non-communicable diseases among older individuals by 25 % within 2025 , in order to counteract the rising wave of population aging and optimize health care resources utilization. 

In this large sample of elderly Italian residents, poor access to health care was a risk factor for poor health outcomes (risk of hospitalization and concurrence of chronic diseases) in all Poisson regression models, highlighting the need of guaranteeing access to health care especially to more vulnerable individuals. The manuscript is well written in all its parts, methods are clearly reported,  statistical analyses are well performed, results are clearly presented, and both introduction and discussion dive in the subject with appropriate references in support. For these reasons, I recommend this manuscript for publication only after minor typo corrections:

-Methods: change people over65-years-of-age with "over 65 years of age".

-In the "health outcomes" part, add a comma between stroke and myocardial infarction. Add periods at the end of the sentences of the list.

-Adjust format of table 1.

Author Response

  • We truly acknowledge the appreciation by the Reviewer 2 and hope that this newly edited version is even more appreciated.

  • Of course, we implemented all three suggestions for changes and, as we report in detail to the other Reviewer, applied several improvement to the article overall as well as in its specific sections and elements.

Round 2

Reviewer 1 Report

The title is now appropriate.

The introduction is not much changes. It includes a lot of very general material about health and access to services, and insufficient references to material about older people, reasons for poor service access and outcomes related to these.

While there is some improvement to the table formatting, result write up is little change and results continue to be difficult to find in amongst the repetition of details of results that are already reported in the tables.

Changes to the Discussion and Conclusion are relatively minor. See issues in my earlier report.
